# Predictors of Symptomatic Venous Thromboembolism in Patients with Soft Tissue Sarcoma in the Lower Extremity

**DOI:** 10.3390/cancers15010315

**Published:** 2023-01-03

**Authors:** Pramod N. Kamalapathy, Adam Kline, Hannah Hollow, Kevin Raskin, Joseph H. Schwab, Santiago Lozano-Calderón

**Affiliations:** Department of Orthopaedic Surgery, Massachusetts General Hospital, Boston, MA 02114, USA

**Keywords:** orthopedic surgery, sarcoma, symptomatic venous thromboembolism (SVTE), DVT prophylaxis

## Abstract

**Simple Summary:**

28 patients, or 4.36%, were diagnosed with venous thromboembolism after soft tissue sarcoma surgery. The most significant risk factors for this complication were pre-operative (PTT) partial thromboplastin time, post-operative PTT, post-op chemotherapy, metastasis at diagnosis, additional surgery for metastasis or local recurrence, and tumor size larger than 10 cm. Risk of wound complications and infection increased in those who received prophylaxis medications.

**Abstract:**

Orthopedic surgery and soft-tissue sarcoma (STS) both independently increase the risk of developing symptomatic venous thromboembolic events (SVTE), but there are no established risk factors or guidelines for how to prophylactically treat patients with STS undergoing surgery. The objectives of this study were to (1) identify the prevalence of SVTE in patients undergoing STS surgery, (2) identify risk factors for SVTE, and (3) determine the risk of wound complications associated with VTE prophylaxis. This retrospective study was conducted in a tertiary level, academic hospital. A total of 642 patients were treated for soft-tissue sarcoma in the lower extremity with follow up for at least 90 days for the development of SVTE such as deep venous thrombosis and pulmonary embolism. Multivariate logistic regression was used to identify predictors for these events by controlling for patient characteristics, surgical characteristics, and treatment variables, with significance held at *p* < 0.05. Twenty eight patients (4.36%) were diagnosed with SVTE. Multivariate analysis found six significant predictors ordered based on standardized coefficients: pre-operative (PTT) partial thromboplastin time (*p* < 0.001), post-operative PTT (*p* = 0.010), post-op chemotherapy (*p* = 0.013), metastasis at diagnosis (*p* = 0.025), additional surgery for metastasis or local recurrence (*p* = 0.004), and tumor size larger than 10 cm (*p* < 0.001). The risk of wound complications (*p* = 0.04) and infection (*p* = 0.017) increased significantly in patients who received chemical prophylaxis. Our study identifies risk factors for patients at increased risk of developing VTE. Further prospective research is necessary to identify which protocols would be beneficial in preventing SVTE in high-risk patients with a low profile of wound complications.

## 1. Introduction

Affecting nearly 350,000–600,000 Americans annually, symptomatic venous thromboembolism (SVTE) represents one of the most common preventable causes of hospital deaths [1,2,3,4]. Venous thromboembolism (VTE) is a blood clot arising in a deep vein, including both deep vein thrombosis (DVT) and pulmonary embolism (PE). Cancer and orthopedic surgery have been independently identified as risk factors for SVTE [5,6,7,8,9]. The literature reports that the incidence of SVTE in this patient population is between 1.4 and 21% with the use of mechanical and/or chemical prophylaxis [8,10,11,12,13]. Through the expression of thrombin and the release of microparticles that influence the solidity of the blood, cancer cells increase coagulation [14,15] and subsequent platelet formation, which is then thought to facilitate the metastatic process [16].

Although all types of surgery increase the risk of clotting and therefore SVTE, orthopedic surgery involves many prothrombotic processes such as coagulation activation from tissue and bone injury; venous injuries; and long periods of immobilization, which further increase the risk and occurrence of SVTE [17]. Thus, this population with STS are at increased risk of undergoing a postoperative thromboembolic event compared to patients after a soft-tissue orthopedic procedure with non-oncologic characteristics [8].

There are currently a few existing studies that evaluate the unique risk factors of an STS patient undergoing orthopedic surgery [18,19,20]. However, these studies are limited by their older case series, small sample sizes, and reliance on national databases. The American Academy of Orthopaedic Surgeons (AAOS) guidelines indicate there is a lack of evidence on the predictors of SVTE in orthopedic oncology patients undergoing surgery, resulting in unclear protocols for post-surgical prophylaxis treatment [11]. Prophylactic anticoagulation medication may help to reduce SVTE and the resulting morbidity and mortality; however, it can lead to surgical bed bleeding, hematoma, and wound complications [21,22]. Identifying SVTE predictors may be extremely beneficial for clarifying and understanding a proper treatment regimen for patients at risk, while helping surgeons to minimize the risks related to anticoagulation. The objectives of this study were to (1) identify the prevalence of SVTE in patients undergoing soft tissue sarcoma surgery, (2) identify risk factors for SVTE in this population, and (3) determine the risk of wound complications associated with VTE prophylaxis.

## 2. Materials and Methods

This retrospective study was conducted at a single large referral institution in order to assess the predictors of soft-tissue high-grade sarcoma of the lower extremity. A total of 642 patients were found using a research patient data registry search that included patients older than 18 years of age who have been surgically treated at our institution for soft tissue sarcomas from January 1992 to December 2017. All tumors were microscopically confirmed to be high-grade soft-tissue sarcomas in the lower extremity, excluding the pelvis, buttocks, and genitals. Exclusion criteria included patients with primary bone sarcomas extending and/or metastasizing to soft tissue or those without follow-up for at least 90 days after surgery. The average age at diagnosis of the population was 53.2 years old, and 56% of patients were male. Body mass index was similar, at 28.1 in no-SVTE patients and 28.5 in SVTE patients (Table 1).

### 2.1. Outcomes of Interest

The outcome event was defined as a radiographically (CT scan or ultrasound) confirmed clinically symptomatic DVT or PE within 90 days of the index surgery. Within the scope of this research, patients were followed up at least at two weeks, six weeks, or three months after surgery. Imaging was only obtained in patients with clinical presentation of SVTE. Routine imaging was not used, as previous literature has now recognized an increase in the prevalence of thromboembolic events if asymptomatic patients are included [19]. The treatment of asymptomatic patients is also controversial [10].

In our practice, DVT prophylaxis is given to patients for at least four weeks. If the patient remains immobile, DVT prophylaxis extends to six weeks or as needed. The protocol period of immobilization after surgery is two days bed-rest after resection if preoperative radiation therapy is given and the condition of the soft tissues is suboptimal; or five days of bed rest with limited dangling if preoperative radiation therapy is given and a rotational flap ± skin graft is used for closure (there is no restriction if radiation was not used preoperatively); or seven days of bed rest and limited dangling, independent of the use of radiation therapy, with a free flap was used for closure.

Information on variables such as patient characteristics, details about the outcome event, and tumor characteristics; treatment variables; preoperative and postoperative clinical variables; and any complications was collected. Tumor site, gross histology, tumor size, and depth were confirmed by the pathology reports and operation notes. Our institution determines grade based on a 1–3 scale, and stage was determined using the American Joint Committee on Cancer staging system (AJCC), which utilizes tumor size, metastases, and grade [23,24]. If a patient presented with an STS tumor in the upper extremity that metastasized to the lower extremity, metastasis was recorded as presenting at diagnosis. Wound complications were defined as a broad category that included infections and conditions such as wound dehiscence, delayed wound healing, local thrombosis, and soft-tissue reconstruction failures. Additional procedures were confirmed by additional operative notes for irrigation and debridement, new soft-tissue reconstructions with grafts or flaps, revisions of these, and other complications. Pre-operative and post-operative variables within a week before and after the surgery were also used. Follow-up varied between patients; while some patients exhibited symptoms of concern and were assessed for DVT quickly, others were only assessed for DVT during regular scheduled follow-ups based on the guidelines stated above.

### 2.2. Statistical Analysis

Patients with SVTE events were compared to those without them to identify any potential predictors of SVTE. All variables were assessed with a multivariate regression model by controlling for patient characteristics, surgical characteristics, and treatment variables. Odds ratios and 95% confidence intervals were reported for all outcomes. Due to the large nature of this retrospective study, there were missing data for a small percentage of patients. Each analysis included only those patients with the variable of interest available. Statistical significance was held at *p* < 0.05. STATA 15 by StataCorp (College Station, TX, USA: StataCorp LLC) was used for all statistical analyses.

## 3. Results

A total of 28 patients (4.36%) out of 642 were diagnosed with SVTE within 90 days of surgery (27 DVT and 1 PE). The average age at diagnosis of the population was 53.2 years old. Fifty six percent of patients were male. There were no differences in the age of patients, body mass index, or smoking status between no-SVTE patients and SVTE patients.

A multivariate logistic model found that tumor size larger than 10 cm (OR 2.41, 95% CI 1.07–5.21), post-surgical chemotherapy (OR 2.98, 95% CI 1.35–6.42), pre-op PTT (OR 0.77, 95% CI 0.68–0.89), post-op PTT (OR 0.91, 95% CI 0.75–0.98), metastasis at diagnosis (OR 3.18, 95% CI 1.11–8.59), and additional surgery for metastasis or local recurrence (OR 2.89, 95% CI 1.24–6.97) were significant predictors of SVTE (Table 2, Table 3 and Table 4).

Of the 642 people, 484 received at least one VTE prophylactic agent. The most common agent was low-molecular-weight heparin (LMWH), which was given to 244 people (Table 3). Patients had no differences in VTE rates based on their chemoprophylaxis (*p* > 0.05). Moreover, patients that received chemoprophylaxis were associated with increased risk of wound complications (OR 1.20, CI 1.01–1.43, *p* = 0.04) and infection (OR 1.24, CI 1.04–1.48, *p* = 0.017). However, no specific chemical prophylaxis was found to be associated with increased wound complication risk (Table 5).

## 4. Discussion

Orthopedic surgery and cancer are both independently associated with an increased risk of developing SVTE [25]. Currently, there are no guidelines that take into account the unique risk factors of this population for prescribing DVT prophylaxis. This study was designed to identify potential predictors of SVTE in STS patients and complications associated with prophylaxis treatments. This study identified the prevalence rate of SVTE following soft tissue sarcoma surgery to be 4.36%. Six significant predictors—post-op PTT, pre-op PTT, post-op chemotherapy, metastasis at diagnosis, additional surgery for metastasis or local recurrence, and tumor size larger than 10 cm—were found to associated with an increased risk of developing SVTE after surgery while adjusting for patient characteristics, tumor characteristics, treatments, and laboratory values.

The prevalence of SVTE in our cohort was slightly lower than that reported in the literature. The percentage of reported SVTE incidence rates in orthopedic surgery varies considerably, ranging from 0.6 to 21% [8,10,11,12,13]. One possible reason for this considerable range could be attributed to the lack of standard protocol used to diagnose SVTE in the published literature. This variation can also be attributed to the variation in the diagnosis of subclinical VTE, which often goes undiagnosed. Our specific cohort did not include imaging studies in asymptomatic patients, reflecting the current clinical practice in which only symptomatic patients are tested. Patients with clinical concern of DVT or PE underwent further imaging, while those with ailments such as unilateral swelling underwent lower-extremity DVT ultrasound. Yet, this is one of the largest cohorts of patients with soft-tissue sarcoma followed to date.

The increase in the risk of SVTE due to metastasis is in line with the idea that the diffuse nature of the tumor leads to hypercoagulability, increasing the risk of thrombosis [26,27]. Metastatic cancers are usually known to be larger and to release more procoagulant factors, requiring a multitude of treatments and leading to shorter survival times. One such treatment, postoperative chemotherapy, is also a significant predictor of SVTE and major bleeding complications [5,28]. Postoperative chemotherapy puts stress on the body and exacerbates any irregular clotting abnormalities [28]. Similarly, additional surgery due to local recurrence or metastasis, which itself is known to be positively predictive of developing DVTs, is significantly correlated with SVTEs, as surgery increases the risk of immobilization and other prothrombic factors. Radiation therapy was likewise expected to be a predictor of SVTE, but it was not found to be significantly associated [26,29]. Radiation therapy has been shown to cause endothelial prothrombotic response, influencing the thrombomodulin complex and various cytokines [10,30] in oncologic patients, but its effects on microcirculation is one possible reason why this discrepancy exists [31].

Our study found that preoperative and postoperative PTT were significantly associated with an increased risk of DVT. Activated PTT is a measure of intrinsic coagulation pathways, meaning PTT levels can be used to measure the rate of coagulation. A low level of PTT indicates a procoagulation tendency because of a greater number of clotting proteins; therefore, it follows that low PTT levels are predictive factors for DVT [32].

Our study did not find any specific chemoprophylaxis associated with significantly decreased risk of SVTE. Currently, AAOS does not have a standard recommendation of DVT prophylaxis for patients undergoing soft-tissue sarcoma patients. Regarding elective hip and knee arthroplasty, AAOS only has evidence sufficient to screen patients with a previous history of SVTE as a risk factor [33]. The American College of Clinical Pharamacology (ACCP) recommends some form of chemical prophylaxis for all patients undergoing major orthopedic surgery; however, there is no standard guideline for which prophylaxis can be employed [34].

Heparin was used as the standard for DVT prophylaxis in the early 1920s until the introduction of warfarin in 1948 [35]. In most cases, heparin was followed by warfarin as a treatment regimen. Eventually, LMWH was issued to alleviate the need to consistently monitor the patient [21]. Many studies identifying risk factors have low statistical power due to the rare occurrence of SVTE and the lack of large data collection in this specific group of patients [36]. Levine et al. demonstrated that LMWH is an equally effective alternative to the unfractionated heparin delivered in the hospital [22]. Singh et al. found that the incidence of DVT in patients undergoing orthopedic oncology lower-limb surgery was low even without prophylaxis, but noted that further investigation with larger sample sizes was necessary [36].

All patients at our institution receive mechanical prophylaxis, either compression stockings or intermittent pneumatic compression devices. In total, 484 of the patients at our institution received at least one prophylactic treatment (Table 3). Chemical prophylaxis is positively associated with wound complications and infection, but no specific prophylactic agent was found to lead to significantly increased risk [36]. This suggests that patients might be over anticoagulated and placed at risk of hematoma formation, with subsequent wound complications and infections. This compounds the necessity of analyzing risk factors for developing SVTE in order to prescribe a patient the proper treatment and minimize their overall complications.

This study had a number of limitations. First, due to its retrospective design, there was incomplete information available for different patients. The large cohort size reduced the effect of the loss of data, as we excluded patients with missing data in each of the independent calculations to reduce concerns. Second, assessing for SVTE was not routine unless the patient was symptomatic and treated at our institution, explaining the lower rate of SVTE recorded in this study compared to in the literature. A prospective study would be helpful to further evaluate the conclusions of this study. However, our design reflects the current standard clinical practice or screening. Third, the study was limited to our institution. This could potentially lead to a more homogenous patient population and lack of generalizability. Our large referral cancer center treats a wide variety of patients, making this limitation less of a concern. Fourth, the present paper attempts to provide a comprehensive analysis of the factors that could contribute to venous thromboembolism postoperatively; however, there are a number of factors, such as prothrombotic agents or previous thromboembolism, that could not be analyzed due to the limited sample size or power of the study. Despite these challenges, the conclusions reached in this study provide clinicians valuable information about orthopedic oncology patients with soft-tissue sarcomas of the lower extremities and how to assess their risk for SVTE.

## 5. Conclusions

Six variables were found to be significant predictors of SVTE in orthopedic oncology patients undergoing surgery: tumor size greater than 10 cm, metastasis of tumor at diagnosis, postoperative chemotherapy, preoperative and postoperative PTT, and additional surgeries. Surgeons and healthcare professionals could minimize the risk of developing SVTE for STS patients by actively following patients with increased risk factors and reducing complications associated with their surgery and recovery. Thrombophylaxis is a gray area in cancer patients, with further prospective studies being required in order to determine which protocols in high-risk patients would be beneficial in preventing SVTE with a low profile of complications in terms of wound healing, postoperative hematoma, and infections.

## Figures and Tables

**Table 1 cancers-15-00315-t001:** Patient characteristics.

Variable	No VTE (614)	VTE (28)	Odds Ratio (95% CI)	*p*-Value
Age at diagnosis	53.2 (40.8–66)	53.4 (44–60.8)	0.99 (0.97–1.02)	0.984
BMI ^a^	28.1 (23–31.4)	28.5 (26.1–30.7)	1.00 (0.93–1.06)	0.918
Sex				
Male	344 (94.8%)	19 (5.2%)	Reference	
Female	270 (96.8%)	9 (3.2%)	2.65 (1.02–8.21)	0.07
Smoking status ^a^				
Never smoked	273 (95.8%)	12 (4.2%)	Reference	
Current smoker	65 (94.2%)	4 (5.8%)	2.22 (0.83–6.32)	0.114
Quit	195 (94.2%)	12 (5.8%)	2.31 (0.58–7.98)	0.196

^a^: Age at diagnosis is available for 606 patients; smoking status is available for 564 people.

**Table 2 cancers-15-00315-t002:** Tumor characteristics.

Variable	No VTE (614)	VTE (28)	Odds Ratio (95% CI)	*p*-Value
Histology				
Undifferentiated/general	91	4	Reference	
Leiomyosarcoma	150	2	0.30 (0.04–1.57)	0.170
Fibrosarcoma	169	5	0.67 (0.17–2.76)	0.556
Angiosarcoma	47	0	-	0.992
Liposarcoma	50	5	2.23 (0.56–9.36)	0.247
Malignant peripheral nerve Sheath tumor	26	0	-	0.996
Rhabdomyosarcoma	10	2	4.13 (0.53–2.39)	0.124
Synovial sarcoma	71	10	3.15 (1.01–1.18)	0.060
Site				
Thigh	378 (95.2%)	19 (4.8%)	Reference	
Leg	198 (97.1%)	6 (2.9%)	0.94 (0.32–2.42)	0.901
Foot	35 (9.2%)	3 (90.8%)	1.92 (0.28–7.78)	0.416
Grade ^a^				
1/3	53 (94.6%)	3 (5.4%)	Reference	
2/3	202 (96.2%)	8 (3.8%)	0.50 (0.13–2.48)	0.348
3/3	182 (95.3%)	9 (4.7%)	0.51 (0.14–2.68)	0.408
1–2/3	26 (96.3%)	1 (3.7%)	-	0.988
2–3/3	128 (94.8%)	7 (5.2%)	0.70 (0.16–3.60)	0.637
Stage ^a^				
I	54 (94.7%)	3 (5.3%)	Reference	
II	206 (97.2%)	6 (2.8%)		0.247
IIIA	156 (94.5%)	9 (5.5%)		0.993
IIIB	100 (97.1%)	3 (2.9%)		0.
IV	76 (91.6%)	7 (8.4%)		0.989
**Dimension larger than 10 cm ^a^**	**279 (94.9%)**	**15 (5.1%)**	**2.41 (1.07–5.21)**	**<0.001**
Vascular Invasion ^a^	55 (94.8%)	3 (5.2%)	1.53 (0.35–4.84)	0.515
**Metastasis at Diagnosis ^a^**	**67 (89.3%)**	**8 (10.7%)**	**3.18 (1.11–8.59)**	**0.025**

^a^: Grade and stage are available for 619 people, dimension for 623, vascular invasion for 604, and metastasis at diagnosis for 500 people. Significant values are bolded.

**Table 3 cancers-15-00315-t003:** Surgery and other treatment.

Variable	No VTE (614)	VTE (28)	Odds Ratio (95% CI)	*p*-Value (Multivariate)
Surgery				
Operative time	2.0 (1.5–2.6)	2.7 (2.1–3.3)	1.11 (0.72–2.87)	0.387
Vascular injury	10 (1.4%)	0 (0%)	-	-
Positive margin ^a^	113 (94.2%)	7 (5.8%)	1.64 (0.57–4.38)	0.483
Blood loss ^a^	374.1 (50–350)	777.7 (125–778)	1.22 (0.74–3.20)	0.519
Reconstruction	396 (97.3%)	11 (2.7%)	1.34 (0.84–2.45)	0.221
Graft ^a^	112 (96.6%)	4 (3.4%)	1.00 (0.28–2.79)	0.982
Tourniquet use	121 (97.6%)	3 (2.4%)	1.2 (0.43–4.10)	0.456
Chemotherapy				
Pre-op ^a^	173 (94%)	11 (6%)	1.50 (0.59–3.68)	0.278
**Post-op ^a^**	**137 (91.3%)**	**13 (8.7%)**	**2.98 (1.35–6.42)**	**0.013**
Radiation				
Pre-op ^a^	373 (95.6%)	17 (4.4%)	1.33 (0.54–3.56)	0.838
Post-op ^a^	138 (95.2%)	7 (4.8%)	1.13 (0.44–2.59)	0.733
VTE prophylaxis				
None	156 (98.7%)	2 (1.3%)	Reference	Reference
ASA	51 (98.1%)	1 (1.9%)	8.37 (0.36–9.97)	0.099
LMWH	244 (93.8%)	16 (6.2%)	4.54 (0.99–2.92)	0.057
Warfarin	149 (94.3%)	9 (5.7%)	3.49 (0.73–2.48)	0.142
Multiple	14 (100%)	0	-	0.994

^a^: Margin information is available for 629 patients, blood loss for 619, graft for 620, pre-op chemo for 603, pre-op radiation for 631, and post-op radiation information for 625. Significant values are bolded.

**Table 4 cancers-15-00315-t004:** Pre-operative and post-operative blood values and complications.

Variable	No VTE (614)	VTE (28)	Odds Ratio (95% CI)	*p*-Value (Multivariate)
Pre-Op				
**Partial thromboplastin ^a^**	30.39 (24.9–30.25)	27.11 (24.5–29.2)	**0.77 (0.68–0.89)**	**<0.001**
PT(INR) ^a^	1.08 (1–1.1)	1.119 (1–1.1)	1.41 (0.20–5.24)	0.648
WBC ^a^	7.40 (5.56–8.46)	7.74 (5.15–8.35)	0.978 (0.83–1.08)	0.754
PLT ^a^	284.60 (206–332)	298.4 (229–298)	1.00 (0.98–1.10)	0.734
HGB ^a^	13.11 (11.3–14.6)	13.47 (10.95–14.8)	1.03 (0.91–1.14)	0.375
**Post Op**				
**Partial thromboplastin ^a^**	38.62 (25.9–39.2)	33.58 (25.7–33.6)	**0.91 (0.75–0.98)**	**0.010**
PT(INR) ^a^	1.17 (1.01–1.20)	1.13 (1.1–1.2)	0.36 (0.01–4.24)	0.228
WBC ^a^	9.21 (7–10.7)	9.94 (7.1–13.78)	1.07 (0.79–1.14)	1.304
Complications				
Infection	116 (92.8%)	9 (7.2%)	1.21 (0.41–3.11)	0.216
Wound Complication	120 (92.3%)	10 (7.7%)	2.25 (1.07–5.21)	0.124
**Additional Surgery for metastasis or local recurrence**	**141 (92.2%)**	**12 (7.8%)**	**2.89 (1.24–6.97)**	**0.004**

^a^: Pre-op values: PTT values is available for 529 patients, PT/INR (383 patients), PLT (573 people), WBC and HGB (576 people), and glucose (456 people). Post-op information: PTT is available for 506 patients, PT/INR (540 people), WBC (509 people). Significant values are bolded.

**Table 5 cancers-15-00315-t005:** DVT prophylaxis and wound complication risk.

Variable	*p*-Value (Multivariate)	Odds Ratio	CI Interval
None	Reference		
Aspirin	0.098	2.49	0.1–42.0
Warfarin	0.089	4.25	0.38–9.46
LMWH	0.066	7.68	0.22–15.2
Multiple treatments	0.078	4.60	0.4–10.3

## Data Availability

Data is unavailable to be made public due to privacy or ethical restrictions, but can be considered based on request to corresponding author.

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
