# Peer review of "Predictors of Symptomatic Venous Thromboembolism in Patients with Soft Tissue Sarcoma in the Lower Extremity"

_cancers, 2023, doi:10.3390/cancers15010315_

Round 1

Reviewer 1 Report

In this manuscript the authors provide results from a retrospective study assessing clinical and laboratory predictors for the development of symptomatic postoperative thromboembolic events (SVTE) such as deep vein thrombosis (DVT) or pulmonary embolism (PE). In this study 642 patients with high-grade sarcoma of the lower limb in the time from 1992-2017 were included. Using multivariant regression, the authors present six significant predictors for the development of a symptomatic thromboembolic event after surgery for high grade sarcoma. These predictors include tumor size greater than 10 cm, metastasis at diagnosis, postoperative chemotherapy, preoperative and postoperative PTT as well as additional surgeries for metastasis or local recurrence. Moreover, patients receiving anticoagulative chemoprophylaxis were found to have a higher risk of wound complications, however this risk wasn’t associated with a specific VTE prophylactic agent.

The presented data is nicely processed and presented in a comprehensive manner. With this manuscript the authors provide valuable information, which can be taken into consideration when evaluation the risk of postoperative thromboembolic events in patient with soft tissue sarcoma of the lower limb. In principle the presented data seems to be both relevant and valid, there are however some aspects and concerns which need to be assessed further: 

1.     As the authors correctly describe in the discussion, PTT is a measure of intrinsic coagulation activity. Low PTT levels reflect a higher coagulative activity, which is why it makes sense that low PTT levels would resolve in a higher risk of venous thromboembolism. PTT being a predictor for SVTE is therefore not surprising as it is a known indicator of coagulation in literature, as well as in clinical day to day work. 
In Table 4 however the No-VTE groups has lower PTT levels (27.11 pre-op, 33,58 post-op) than the VTE group (30,39 pre-op, 38,62 post-op). Following the logic above the No-VTE group with low PTT levels should have a higher coagulation activity, still the thromboses happened in the other group. This inconsistency needs to be either corrected, if the data in the table is false, or discussed further, if this is in fact the case.

2.     Causes of thromboembolic events are often multidimensional. Has there been an assessment of medication especially prothrombogenic drugs or oral anticoagulation preoperatively? Is data on previous thromboembolic events available in the cohort? If so, these factors should be controlled for like it has been done for age, body mass index or smoking.

3.     Also, some of the stated predictors could influence one another. E.g. is there data on why the patients received postoperative chemotherapy, which has been identified as an independent risk factor for the development of an SVTE? How many of the patients receiving postoperative chemotherapy had metastases at time of diagnosis? Is there a way to control for confounding effects within the six newly found predictors? If this has already been done, it should be further highlighted in the statistics section or mentioned again in the discussion.

Author Response

Author 1:

In this manuscript the authors provide results from a retrospective study assessing clinical and laboratory predictors for the development of symptomatic postoperative thromboembolic events (SVTE) such as deep vein thrombosis (DVT) or pulmonary embolism (PE). In this study 642 patients with high-grade sarcoma of the lower limb in the time from 1992-2017 were included. Using multivariant regression, the authors present six significant predictors for the development of a symptomatic thromboembolic event after surgery for high grade sarcoma. These predictors include tumor size greater than 10 cm, metastasis at diagnosis, postoperative chemotherapy, preoperative and postoperative PTT as well as additional surgeries for metastasis or local recurrence. Moreover, patients receiving anticoagulative chemoprophylaxis were found to have a higher risk of wound complications, however this risk wasn’t associated with a specific VTE prophylactic agent.

The presented data is nicely processed and presented in a comprehensive manner. With this manuscript the authors provide valuable information, which can be taken into consideration when evaluation the risk of postoperative thromboembolic events in patient with soft tissue sarcoma of the lower limb. In principle the presented data seems to be both relevant and valid, there are however some aspects and concerns which need to be assessed further: 

  1. As the authors correctly describe in the discussion, PTT is a measure of intrinsic coagulation activity. Low PTT levels reflect a higher coagulative activity, which is why it makes sense that low PTT levels would resolve in a higher risk of venous thromboembolism. PTT being a predictor for SVTE is therefore not surprising as it is a known indicator of coagulation in literature, as well as in clinical day to day work. 
    In Table 4however the No-VTE groups has lower PTT levels (27.11 pre-op, 33,58 post-op) than the VTE group (30,39 pre-op, 38,62 post-op). Following the logic above the No-VTE group with low PTT levels should have a higher coagulation activity, still the thromboses happened in the other group. This inconsistency needs to be either corrected, if the data in the table is false, or discussed further, if this is in fact the case. 

Thank you for your feedback. We have adjusted the inconsistency in transferring the tables. The lower PTT levels are in fact associated with the VTE groups and have been adjusted appropriately in the tables and throughout the manuscript.

  1. Causes of thromboembolic events are often multidimensional. Has there been an assessment of medication especially prothrombogenic drugs or oral anticoagulation preoperatively? Is data on previous thromboembolic events available in the cohort? If so, these factors should be controlled for like it has been done for age, body mass index or smoking.

Thank you for these suggestions. We had previously assessed these two factors; Unfortunately, due to the limitations of the dataset, there were not enough patients or power to adequately analyze the impact of prothrombotic drugs or oral anticoagulation. We have added that to the limitations of the manuscript.

On line 222-225 we have added to the limitations: “Fourth, the present paper attempts to provide a comprehensive analysis of factors that could contribute to venous thromboembolism postoperatively; however, there are a number of factors, prothrombotic agents or previous thromboembolism, that could not be analyzed due to limited sample size or power of the study.”

  1. Also, some of the stated predictors could influence one another. E.g. is there data on why the patients received postoperative chemotherapy, which has been identified as an independent risk factor for the development of an SVTE? How many of the patients receiving postoperative chemotherapy had metastases at time of diagnosis? Is there a way to control for confounding effects within the six newly found predictors? If this has already been done, it should be further highlighted in the statistics section or mentioned again in the discussion.

An extensive multivariate analysis was undertaken to attempt to control patient demographics, tumor presentation, and treatments in order to identify significant predictors. A total of 20 patients out of 137 patients that underwent postoperative chemotherapy had metastasis at the time of diagnosis. We have added to lines 155-156:

“Six significant predictors-- post-op PTT, pre-op PTT, post-op chemotherapy, metastasis at diagnosis, additional surgery for metastasis or local recurrence, and tumor size larger than 10cm--were found to associated with an increased risk of developing SVTE after surgery while adjusting for patient characteristics, tumor characteristics, treatments, and laboratory values.”

Reviewer 2 Report

Thank you for the opportunity to review this paper. I think it is well written and a pertinent topic for the Orthopaedic Oncology community as a whole. One of the most important findings in this large cohort study is identifying a symptomatic VTE rate of ~4.4%.

However, I do think there are some considerations in the methodology that necessitate improvement. I think the primary limitation is one that is difficult to remedy: the inherently low incidence of symptomatic VTE (quoted in this study at 4%, which is higher than the typical population certainly). With only 28 patients in the comparison group, I think that any significant comparisons will be inherently limited. This is a problem inherent to the study subject and is a limitation not easily overcome. Nonetheless, these are interesting findings.

The claim that there are no studies examining VTE in STS is not exactly correct. A review of the literature identifies the following studies that have previously investigated this subject matter. While I think the present study has more sound methodology, the following studies have investigated this subject matter already:

Shantakumar et al. Older soft tissue sarcoma patients experience increased rates of venous thromboembolic events: a retrospective cohort study of SEER-Medicare data

https://pubmed.ncbi.nlm.nih.gov/26213607/

Alcindor et al. Venous Thromboembolism in Patients with Sarcoma: A Retrospective Study

https://www.ncbi.nlm.nih.gov/pmc/articles/PMC6519765/.

Krzyzaniak et al.  Venous thromboembolism rates in patients with bone and soft tissue sarcoma of the extremities following surgical resection: A systematic review

https://pubmed.ncbi.nlm.nih.gov/33866561/

And others. I would suggest the authors discuss how their study differs significantly from these and include references to incorporate more of the existing literature base.

Additionally, the authors quote studies regarding carcinoma VTE rate. There is some controversy over sarcoma VTE rates. While generally thought to be higher, whether sarcoma itself is pro-thrombotic is not well established. I would ask the authors to elaborate on this point.

Methods:

Section 2.1 Please elaborate upon what is specifically meant by clinically symptomatic DVT or PE? Presumably the authors had an algorithm for what warranted work up.

Is this period of immobilization really used so ubiquitously? Ours and many other institutions are pushing more and more towards early mobilization. Certainly a need for soft tissue coverage may change some of this protocol, but I would be surprised that a small superficial high grade tumor would require a period of bed rest. If this is the case, I would ask the authors to highlight this as a limitation.

Please provide reference for grading schema used.

One of the inherent challenges of this study will be that only 28 patients had a symptomatic VTE. Was a post-hoc power analysis performed for this study for the outcomes of interest? How do you know that you are not missing significant factors simply from a power perspective alone?

Would you mind please elaborating on how the multivariable analysis was performed? Was it ONLY controlled for with age BMI and smoking status?

Results

Why is there a difference in patients with Stage IV disease and metastatic disease at presentation?

Some of these findings are intuitive. It makes sense that a larger tumor would dictate a greater dissection and potentially a greater risk of and need for soft tissue coverage. But it seems to me that one of the most significant factors that would be predictive of VTE would be the postoperative immobilization protocol. If you get a free flap and are stuck in bed for a week, your risk of VTE will be inherently high. Additionally, what about operative time? Vascular injury or reconstruction? Superficial or deep location? I think these would be important factors to consider. Additionally there is probably some inherent confounding there... would it not make sense to control for this in the multivariable analysis?

For the tables, are these means/ medians? SD or IQR?

What does reconstruction mean? Soft tissue reconstruction?

I am having trouble understanding how these mean PTT values translate clinically. Our normal PTT is between 25 to 35. So is this really a significant finding? Additionally, why would PTT be higher in the VTE group? It might make more sense to identify a groups below normal, wnl, and above normal?

It would be helpful to understand indications for or against anticoagulation? I think this is necessary to better interpret these results. Overall I would be reticent to NOT provide patients with anticoagulation after a sarcoma resection of the LE, but presumably a small superficial sarcoma closed primarily might be considered for no anticoagulation while I would assume that a large deep sarcoma would typically receive anticoagulation. But would also be at an increased risk of wound complications etc. Additionally, any consideration of wound complications must also consider preoperative radiation therapy?

What were the additional surgeries performed? Where these all performed within 90 days? That would be surprising to have a LR that quickly. Were these concurrent metastectomies, so presumably the same cohort of patients with known existing metastatic disease?

I think given the above, the multivariable analysis could be reconsidered to help account for some of the more significant confounding variables. I ask the authors to investigate this methodology.

I would ask that the authors address the above concerns in more depth in the discussion.

Author Response

Author 2

  1. Thank you for the opportunity to review this paper. I think it is well written and a pertinent topic for the Orthopaedic Oncology community as a whole. One of the most important findings in this large cohort study is identifying a symptomatic VTE rate of ~4.4%.

However, I do think there are some considerations in the methodology that necessitate improvement. I think the primary limitation is one that is difficult to remedy: the inherently low incidence of symptomatic VTE (quoted in this study at 4%, which is higher than the typical population certainly). With only 28 patients in the comparison group, I think that any significant comparisons will be inherently limited. This is a problem inherent to the study subject and is a limitation not easily overcome. Nonetheless, these are interesting findings.

Thank you. Inherently, we are unable to remedy the situation with the retrospective nature of the study.

  1. The claim that there are no studies examining VTE in STS is not exactly correct. A review of the literature identifies the following studies that have previously investigated this subject matter. While I think the present study has more sound methodology, the following studies have investigated this subject matter already:

Shantakumar et al. Older soft tissue sarcoma patients experience increased rates of venous thromboembolic events: a retrospective cohort study of SEER-Medicare data

https://pubmed.ncbi.nlm.nih.gov/26213607/

Alcindor et al. Venous Thromboembolism in Patients with Sarcoma: A Retrospective Study

https://www.ncbi.nlm.nih.gov/pmc/articles/PMC6519765/.

Krzyzaniak et al.  Venous thromboembolism rates in patients with bone and soft tissue sarcoma of the extremities following surgical resection: A systematic review

https://pubmed.ncbi.nlm.nih.gov/33866561/

And others. I would suggest the authors discuss how their study differs significantly from these and include references to incorporate more of the existing literature base.

Additionally, the authors quote studies regarding carcinoma VTE rate. There is some controversy over sarcoma VTE rates. While generally thought to be higher, whether sarcoma itself is pro-thrombotic is not well established. I would ask the authors to elaborate on this point.

Thank you for pointing out these references. We have added these to our manuscript. Our tertiary care center compiles one of the largest cohorts available for analysis. It is important to understand regional differences between the studies represented in the references above, as well as, increasing the sample sizes available to understand the true VTE risk in this population.

Methods:

Section 2.1 Please elaborate upon what is specifically meant by clinically symptomatic DVT or PE? Presumably the authors had an algorithm for what warranted work up.

Patients with clinical concern for DVT or PE underwent further imaging at routine follow ups. For example, patients with unilateral lower extremity would undergo DVT ultrasound.

Is this period of immobilization really used so ubiquitously? Ours and many other institutions are pushing more and more towards early mobilization. Certainly a need for soft tissue coverage may change some of this protocol, but I would be surprised that a small superficial high grade tumor would require a period of bed rest. If this is the case, I would ask the authors to highlight this as a limitation.

While we certainly also push towards early mobilization, patients undergoing surgical resection can be certainly have difficulty mobilizing for longer periods and inherently are at increased risk of VTE. Often wound risk is guiding our immobilization: two days bed-rest after resection if preoperative radiation therapy is given and the condition of the soft tissues is suboptimal, or five days of bed rest with limited dangling if preoperative radiation therapy is given and a rotational flap plus/minus skin graft is used for closure (there is no restriction if radiation was not used preoperatively, or seven days of bed rest and limited dangling (independent of the use of radiation therapy) is given and a free flap was used for closure.

Please provide reference for grading schema used.

Thank you. We have added the reference.

One of the inherent challenges of this study will be that only 28 patients had a symptomatic VTE. Was a post-hoc power analysis performed for this study for the outcomes of interest? How do you know that you are not missing significant factors simply from a power perspective alone?

We did in fact consider post ad hoc power analysis for the present study given the limitations of the study; however, it has been cited that post hoc power calculations are not useful as it is completely determined by p-value. Furthermore, it does not provide enough information that insignificant results have “low power”.

Would you mind please elaborating on how the multivariable analysis was performed? Was it ONLY controlled for with age BMI and smoking status?

Patients were controlled using all the variables available to ensure that there no confounding significant factors.

Results

Why is there a difference in patients with Stage IV disease and metastatic disease at presentation?

https://www.cancer.org/cancer/soft-tissue-sarcoma/detection-diagnosis-staging/staging.html

Stage 4 was characterized as either spread to nearby lymph nodes or cancer that has spread to nearby lymph nodes or to distant sites (metastatic).

Some of these findings are intuitive. It makes sense that a larger tumor would dictate a greater dissection and potentially a greater risk of and need for soft tissue coverage. But it seems to me that one of the most significant factors that would be predictive of VTE would be the postoperative immobilization protocol. If you get a free flap and are stuck in bed for a week, your risk of VTE will be inherently high. Additionally, what about operative time? Vascular injury or reconstruction? Superficial or deep location? I think these would be important factors to consider. Additionally there is probably some inherent confounding there... would it not make sense to control for this in the multivariable analysis?

We have added operative time and vascular injury as well. These were not found to be significant.

For the tables, are these means/ medians? SD or IQR?

We used SD and IQR as these populations are not normalized

What does reconstruction mean? Soft tissue reconstruction?

Reconstruction refers to soft tissue reconstruction with grafting or flaps. It is described in the methodology on lines 102.

I am having trouble understanding how these mean PTT values translate clinically. Our normal PTT is between 25 to 35. So is this really a significant finding? Additionally, why would PTT be higher in the VTE group? It might make more sense to identify a groups below normal, wnl, and above normal?

We apologize for the mischaracterization. The PTT values were in fact switched in the tables/results and PTT is lower in the VTE group. We have made the appropriate changes throughout the manuscript.  

It would be helpful to understand indications for or against anticoagulation? I think this is necessary to better interpret these results. Overall I would be reticent to NOT provide patients with anticoagulation after a sarcoma resection of the LE, but presumably a small superficial sarcoma closed primarily might be considered for no anticoagulation while I would assume that a large deep sarcoma would typically receive anticoagulation. But would also be at an increased risk of wound complications etc. Additionally, any consideration of wound complications must also consider preoperative radiation therapy?

The wound complications were referenced while accounting for preoperative variables such as radiation therapy. In our practice, DVT prophylaxis is given to patients for at least four weeks. If the patient remains immobile, DVT prophylaxis extends to six weeks or as needed

What were the additional surgeries performed? Where these all performed within 90 days? That would be surprising to have a LR that quickly. Were these concurrent metastectomies, so presumably the same cohort of patients with known existing metastatic disease?

On lines 101-102, it is explained in the methodology: Additional procedures were confirmed by additional operative notes for irrigation and debridement, new soft-tissue reconstructions or revisions of and other complications

I think given the above, the multivariable analysis could be reconsidered to help account for some of the more significant confounding variables. I ask the authors to investigate this methodology.

We have adjusted the manuscript to become more clear that we accounting for all the preoperative variables available.

I would ask that the authors address the above concerns in more depth in the discussion.

Thank you. We hope we have addressed the above concerns.